# Analysis of Running-Related Injuries: The Vienna Study

**DOI:** 10.3390/jcm9020438

**Published:** 2020-02-06

**Authors:** Emir Benca, Stephan Listabarth, Florian K.J. Flock, Eleonore Pablik, Claudia Fischer, Sonja M. Walzer, Ronald Dorotka, Reinhard Windhager, Pejman Ziai

**Affiliations:** 1Department of Orthopedics and Trauma Surgery, Medical University of Vienna, 1090 Vienna, Austria; stephan.listabarth@meduniwien.ac.at (S.L.); sonja.walzer@gmx.at (S.M.W.); reinhard.windhager@meduniwien.ac.at (R.W.); 2Center for Medical Statistics, Informatics, and Intelligent Systems, Medical University of Vienna, 1090 Vienna, Austria; eleonore.pablik@meduniwien.ac.at; 3Department of Health Economics, Center for Public Health, Medical University of Vienna, 1090 Vienna, Austria; claudia.fischer@meduniwien.ac.at; 4Orthopädie-Zentrum Innere Stadt, 1010 Vienna, Austria; r.dorotka@ortho-zentrum.at (R.D.); ziai@sporthomed.at (P.Z.)

**Keywords:** running related overuse injury, running related injury, etiology, injuries, epidemiology

## Abstract

Background: This study aimed to provide an extensive and up-to-date analysis of running-related injuries (RRI) and analyze a broad range of contributing factors for a large heterogeneous and non-selected running population from Central Europe. Methods: Anthropometric, training, footwear, anatomic malalignment, and injury data from 196 injured runners were assessed case-controlled and retrospectively. Univariate and multivariate regression models were developed to identify associated factors for specific injury locations and diagnoses. Results: The majority of patients were female (56%). Three most frequently observed malalignments included varus knee alignment, pelvic obliquity, and patellar squinting. The most common injuries were the patellofemoral pain syndrome (PFPS), the iliotibial band friction syndrome (ITBFS), patellar tendinopathy, spinal overload, and ankle instability. A number of contributing factors were identified. Previous injury history was a contributing factor for knee injuries and ITBFS. Lower training load was reported with a higher incidence of PFPS, while a higher training load was positively associated with injuries of the lower leg. Runners with a higher body mass index (BMI) were at a significantly higher risk for lower back injuries. Conclusions: Running-related injuries are multifactorial associated with a combination of variables including personal data, training load, anatomic malalignments, and injury history. They can furthermore result from a lack of experience/training as well as from overuse. Suffering a specific RRI of high risk could be defined based on individual predispositions and help to induce appropriate training balance.

## 1. Introduction

According to current surveys, running is one of the favorite sports activities across the globe. For Austria as a Central European Country, surveys show the following numbers: 14% of Austria’s population, age 15 and older, are running at least once a week and another 17% are running less than once a week, but still regularly [1]. Unfortunately and despite the well-known health benefits of regular running exercise, running is associated with running-related injuries (RRI) with a yearly incidence of up to 79% [2,3]. Studies suggest 7.7 running-related injuries for recreational runners, and 17.8 for novice runners per 1000 h of running [4], whereas the vast majority of these injuries are related to overuse [5]. Similarly, 17 injuries were reported per 1000 exposure hours in high-school runners [6]. While injuries to runners are rarely severe, they can be frequent and persistent requiring medical treatment associated with treatment costs. These facts illustrate the necessity for research on both the epidemiology and contributing factors to running-related injuries. Since preventive measures would be the most effective way to reduce the individual and the economic burden of RRIs, identifying potential risk factors presents a necessary step.

Interestingly, there has been limited clinical research on RRIs and risk or associated factors in recent years. While the long-lasting popularity of running has not diminished, available research data were published between 1986 [7] and 2014 [8] (on average 16 years ago) based on even older records. As displayed in Table 1, the vast majority of studies were conducted in North America [2,5,6,7,9,10,11] or northern Europe [12,13,14]. Included subjects were samples from specific athletic populations [6,8,12,13] or participants of specific races [7,9,10,15,16,17]. Some authors investigated specific risks factors associated with RRIs as an unspecific single pathology [2,6,7,8,10,11,15,16,17], whereas it is well known that running is associated with dozens of overuse injuries to soft- and hard tissue in all body sites between the lower back and the foot [5]. Furthermore, reported associated risk factors can be intrinsic and/or extrinsic (including training related factors), whereas different RRIs may develop based on the types of errors in the running regime. In recent years, the running footwear industry has undergone a transformation by introducing minimalist footwear promising improved biomechanics leading to potentially lower injury incidence rates. However, in smaller studies, it was shown that running in minimalist footwear appears to increase the likelihood of experiencing an injury [18]. At the same time, there is no evidence to support the consistently advertised distance running shoe [19]. Yet, no study group has investigated the use of different running shoes and the frequency of RRIs so far. Finally, the most extensive study with more than 2000 patients proposed the necessity for a precise measure of weekly running distance and running experience and investigate their effect on injury incidence [9].

The aim of this detailed up-to-date comprehensive retrospective study was to (1) present anthropometric, training as well as injury data of a heterogeneous running population from a Central European capital, (2) investigate if different running footwear categories affect specific injury incidences, and (3) analyze a broad range of potential contributing factors for the most common running RRIs. We hypothesized that different types of footwear (e.g., motion-controlled or minimal running shoes) are associated with specific injuries (second aim) and that a combination of anthropometric data, preexisting malalignments, and training load result in overloading of structures and in a specific running-related injury (third aim).

## 2. Materials and Methods

The patients (injured runners) were recruited at the orthopedics practice Orthopädie-Zentrum Innere Stadt in Vienna, Austria during a three-year period starting November 2013. The practice was chosen for patient recruitment as it was suited in the center of Vienna and within walking distance from various public transport. The practice has also a contract with all national health care insurance schemes, allowing, on the one hand, the majority of patients receiving medical treatment while being covered by their insurance with an insignificant deductible at maximum and on the other hand an objective and irrespective of costs examination for every participant. It is worth mentioning that the national healthcare system does not incorporate gatekeeping, meaning patients do not always have to seek general practitioners for authorizing access to specialty care, hospital care, and diagnostic tests. That is why a visit to an orthopedist specialist does not automatically imply a severe injury in the Austrian health care setting. This ensures a heterogeneous study population of runners.

Patients visited either the practice self-reliantly or they were referred from their general practitioners. Patient examinations were performed by senior orthopaedics and trauma surgeons, who were also specialized in sports medicine. Typically patients were questioned about the history and nature of the injury. This was followed by an anatomical assessment and physical examination. Anatomical assessment included the following measurements: foot malalignments (by inspection classified in pes cavus, pes planus, pes valgus), patellar squinting (by inspection in femoral anteversion), knee malalignments (by inspecting the distance between the lateral/medial femoral epicondyles with the medial/lateral malleoli for genu varum/valgus), pelvic obliquity (by the distance from the anterior superior iliac spine to the medial malleolus on the supine patient >0.5 cm), scoliosis (by inspection and radiographs if curvature exceeds 10°), etc. Appropriate radiological diagnostic modalities were incorporated if presented or as required. Following the examination, all patients were given a questionnaire and brief information on the study participation.

An injury was related to running if the patient had pain or symptoms during or immediately after a running session and was felt to be related to running, the primary clinical diagnosis was significant enough to keep the runner away from the training routine for more than three days, and to seek medical assistance. Criteria for exclusion were age lower than 18 years, a weekly distance less than 7.5 km or less than one hour running per week or missing the patient’s written consent.

A questionnaire including 31 questions was designed to assess all relevant patient data including personal (chronic diseases (diabetes), cortisone intake, highest education (academic degree), and smoking status (smoker/non-smoker)) and contact data (name, e-mail, and telephone number) as well as anthropometric (sex (female/male), age (years), height (cm), weight (kg), handedness (right/left), number of doctor’s visit within the past 12 months, relevant injury history (injury duration)), training load (activity history (years, months), weekly activity (in km and hours), training frequency (sessions per week), pace (min/km or km/h)), training data (running underground (asphalt/cross-country terrain/concrete/grass/track/sand), time of day (morning/afternoon/evening), other concurrent physical activities including frequency (specific sport/activity (sessions/month), stretching-, warming up habits (yes/no), changes in training within the past 30 days (yes/no)), and footwear (brand, model, change within past 30 days). Patients were further requested to give written consent for their anonymized data to be used for research.

Patient characteristics were summarized as mean ± standard deviation (SD). Differences in anthropometric, personal-, and runner’s profile data between sexes were calculated using the Wilcoxon rank-sum test. The association between the factors: sex, age, height, body mass index (BMI), training load, individual training habits (warming up, cooling down, stretching before, and stretching after run), and malalignments (of back, hip, knee, foot, and ankle) with the five most common injury locations and five most commonly diagnosed specific injuries were calculated using univariate logistic regression models. Variables that showed at least a trend (*p* < 0.1) in the univariable model were combined in a multivariable regression model. Stepwise model selection according to the best AIC (Akaike information criterion) was used to eliminate redundant variables from the model. Each injury analysis revealed significant differences to patients, who did not suffer that specific injury but were suffering any other RRI. The authors chose patients who were suffering any other RRI as a control group based on two considerations: (1) In order to recruit an unbiased control group for a large running population the existing literature suggests either reducing the study collective to a specific athletic population [2,6,8] or to include only participants of specific races [7,9,10,15]. Since this study aimed to describe injuries in a heterogeneous running population, it was inevitable to use a dependent variable of runners with a specific injury and compare them with a control group of runners who experienced a different injury. The assumption that the distribution of contributing factors in the aforementioned control group would at least be comparable to that in a group of non-injured runners was also made in the most extensive study on running-related injuries ([5]). (2) A recent meta-analysis showed the injury incidence per 1000 h to be as high as 7.7 in recreational runners and 17.8 in novice runners [4], while the overall incidences for injuries vary from 26% to 92.4% [3]. Based on both, high incidences and high incidence rates, it again seemed plausible and legitimate to compare runners, who suffered a specific injury with other runners, who suffered any other injury. Significance level alpha was set to 0.05 for all tests, without correction for multiple testing. *p*-Values, therefore, have to be interpreted hypotheses generating only. In case of a low number of positively reported variables (e.g., smoking, medication intake), those variables were not presented nor included in the analysis.

## 3. Results

### 3.1. Descriptive

Table 2 provides an overview of the baseline characteristics and differences between the sexes of the included patients. A total of 196 patients were examined and questioned. Eighteen patients had to be excluded due to the above-defined criteria. Finally, 178 patients were included in the data analysis (see Table 2).

The sex ratio of women:men was 1.25. Calculated BMI values classified 8.6% female patients to suffer from underweight (BMI <18.5 kg/m^2^), while 9.7% women and 31.9% men reported values classified as overweight or obesity (BMI >25 kg/m^2^).

Over the 3-year period a peak in presented injuries could have been observed in March (14.6%), the lowest number of injury presentations was observed in August (Figure 1) (missing values: 11.2%).

### 3.2. Training Data

#### 3.2.1. Training Analysis

The analysis of the patients’ training showed that men reported higher mean values for running history and training load (see Table 3 for more details). The majority of questioned patients run rather in the evening (38.2%) than morning (18.0%) or afternoon (14.0%) (missing values: 29.8%). Patients reported following percentage for predominant running surfaces: asphalt (76.4%), cross-country terrain (47.8%), various surfaces (39.9%), concrete (28.7%), grass (18.5%), track (12.4%), and sand (2.8%) (multiple answers were possible; missing values: 0.0%). A significant number of runners practice a warm-up (51.6%) and cool-down routine (44.9%) and stretch before (35.9%) or after the run (83.7%). There were no missing values for those four categories.

Twenty-nine patients (16.2%) have adapted their training routine (technique, frequency, intensity) 30 days prior to the injury (missing values: 1.6%).

#### 3.2.2. Alternative Sports

Most patients (90.4%) participate in alternative sports other than running (in descending order): cycling, fitness training, hiking, skiing, swimming, others, walking, weight lifting, snowboarding, soccer, aerobics, rollerblading, dance, etc. Women perform these sports on average 9.9 and men 8.2 times a month.

### 3.3. Footwear

The patients were asked via questionnaire for the brand and the specific model of their predominantly used running shoes. Interestingly, all but one participant were able to name the brand, but fewer than half (45.4%) were able to specify the exact shoe model. Only 18.8% of patients have changed their running shoes 30 days prior to the injury (missing values: 1.6%).

The analysis of patients using the often promoted corrective function of motion-controlled traditional running shoes (TRS) in runners with pes valgus is displayed in Table 4. Despite the widely spread recommendation, only 39.1% of patients with pes valgus use motion-controlled TRS (Table 5).

### 3.4. Anatomic Malalignments

Figure 2 shows the distribution of the most common anatomic malalignment across sexes and age groups. Due to a significant number of different malalignments and a limited number of patients, the seven most commonly diagnosed malalignments were presented and included in the statistical analysis. Varus knee alignment was seen in 83.1% patients (84.4% female and 81.0% male), pelvic obliquity in 80.8% (83.8% female, 77.2% male), patellar squinting in 52.2% (51.5% female, 53.1% male), pes cavus in 36.5% (30.3% female, 44.3% male), pes planus in 34.8% (37.3% female, 31.6% male), scoliosis in 31.4% (39.3% female, 21.5% male), and pes valgus in 30.3% patients (33.3% female, 26.5% male).

### 3.5. Injury Analysis

The seen injuries were exclusively overuse injuries. There were 178 patients who suffered 244 injuries with 44 specific diagnoses. Forty-five injuries were secondary injuries to the one and the same location. The most common location of injury was the knee (41.2%) followed by the ankle joint (15.0%) and foot (10.6%) (see Table 6 for more details). The five most common injuries were the patellofemoral pain syndrome (PFPS) (13.4%), the iliotibial band friction syndrome (ITBFS) (12.3%), patellar tendinopathy (12.3%), spinal injuries (11.2%), and ankle instability (8.4%) (Figure 3). Further injuries were not included in the analysis due to the low number of cases and low statistical power.

Hundred-seventy-eight patients suffered 244 injuries. Forty-five injuries were secondary injuries to one and the same location.

On average patients suffered their last injury, causing them to fully interrupt their training routine for at least three days, 10.1 ± 17.1 months (range: 0–108 months) before the current injury (missing values 29.9%) and in 67.2% cases complaints were identical to the current injury (missing values: 7.3%).

### 3.6. Univariate Analysis

The authors evaluated the association between the factors: sex, age, height, BMI, activity history, weekly activity hours, weekly mileage, weekly frequency as well as individual training habits (e.g., warming up, stretching), and various anatomic malalignments with the five most common injury locations including most common specific injuries in these locations. To prevent the exclusion of predictors with borderline significance, a *p*-value of 0.100 rather than 0.050 was used for the inclusion of variables in the multivariate model. Table 7 shows all included relationships (*p* < 0.1), revealed from a univariate model, which could be found in the presented sample.

Knee injuries were positively associated with younger age, lower BMI, and previous injury with similar complaints as well as with a series of malalignments (varus knee, patellar squinting, pes planus, and knee malalignment). The patellofemoral pain syndrome was positively associated with lower weekly activity, lower mileage, lower training frequency, further with lower pace as well as with hip and knee malalignment (especially patellar squinting), and with a lower height. The iliotibial band friction syndrome showed the overall highest positive association with previous injury (OR 10.1). However, the large confidence interval also indicates a low level of precision. ITBFS showed also a negative association with BMI. Patients with patellar tendinopathy were younger, showed less often scoliosis, but more often knee malalignments than the other patients.

Injuries to the lower leg were observed in patients with overall higher training load (more weekly hours, higher mileage, higher frequency, and higher pace), but had a negative relationship with malalignment of the knee in general and patellar squinting in particular.

Patients with foot/ankle injuries reported less often a history of similar previous injuries and showed a negative association with knee malalignment (especially with patellar squinting), while ankle instability was positively associated with weekly hours.

Hip/pelvis patients showed a positive association with running history and scoliosis and a negative one with stretching after the run.

Lower back patients (all had a spinal injury) showed the highest proportion of scoliosis (OR 8.282), however with a very large confidence interval. There were further positively related to BMI, but negatively to warming up and the varus knee.

### 3.7. Multivariate Analysis

Table 8 shows the results of the best multivariable model selected with AIC.

Knee injuries were positively associated with lower BMI, and previous injury with similar complaints as well as with malalignments (knee malalignment, especially varus knee, and pes planus). The patellofemoral pain syndrome was positively associated with lower training frequency, knee malalignment, and with lower height. The iliotibial band friction syndrome showed a positive association with previous injury and a negative association with BMI. Patients with patellar tendinopathy had a lower BMI, showed a lower running pace, but more often knee and less often hip malalignments than the other patients.

Injuries to the lower leg were observed in patients with a higher running pace but had a negative relationship with malalignment of the knee.

Patients with injuries to foot/ankle reported less often a history of similar previous injuries and showed a negative association with knee malalignment and pes planus, while ankle instability was positively associated with weekly activity and pes cavus.

Hip/pelvis patients showed a positive association with running experience, scoliosis, and patellar squinting and a negative one with stretching after the run.

Lower back (spinal) injury patients showed a high proportion of scoliosis (OR 12.669) and were positively related to higher BMI, but negatively to the varus knee.

The model equations are presented in Table 9.

## 4. Discussion

The aims of this study were (1) the presentation of a heterogeneous running population with running-related injuries, (2) analysis a broad range of potential contributing factors for most common RRIs and their combinations, and (3) investigation if different running footwear categories affect injury incidences.

Using a detailed questionnaire in combination with malalignment and injury data, a detailed description of a large population with running-related injury was presented. The running population is characterized by its heterogeneity, long running history, and a detailed description of the training data.

The collected data were used to develop univariate and multivariate logistic regression models. In the multivariate regression model, a number of associated variables remained significant. The majority of variables that were statistically significant in the univariate, but not in the multivariate model, affected the training load. The multivariate model could be applied to asses patient-specific injury risk using patient’s characteristics, preexisting malalignments, and training load. Single variables associated with specific injuries in the multivariate logistic regression model are discussed extensively below.

It is important to mention, that single variables associated with a specific injury or injury location must not be regarded as isolated contributing factors. A RRI is the result of multiple associated factors in terms of anthropometrics and malalignments in combination with previous injury history and exposure to certain training loads. None of the sustained RRIs are associated with a single variable only, e.g., higher BMI or knee malalignment. An injury is sustained when the interaction of predisposing factors, positive injury history and a reached threshold in training load becomes significant to cause an injury. Research questions should not focus on the effect of single variables on a specific injury, but on specific injuries as a combination of multiple variables as well as on the weighting of those variables in the injury prediction models. For example, our data showed BMI to be associated with multiple injuries or injuries locations, however, never as predominant regression weight (−0.16 < regression coefficient < 0.24), when compared with other significant variables. On the other hand, the odds to suffer PFPS, patellar tendinopathy or a knee injury in general, are much higher in patients with knee malalignments (1.13 < regression coefficient < 1.47). In other words present knee malalignment will contribute more to suffering a RRI than BMI and, therefore, shows different clinical relevance.

Finally, the patients were not able to provide sufficient data to properly address the question if different running footwear categories modify injury incidences. Interestingly, a relatively low proportion of runners with the pes valgus deformity wears motion-controlled TRS, even though they are generally recommended for these specific runners.

### 4.1. Injury Summary

Our analysis showed that the knee, followed by foot/ankle, lower leg, and hip/pelvis were, in decreasing order, the most commonly affected anatomical locations, which was in line with the literature [5,9,11]. Three most commonly diagnosed pathologies were the patellofemoral pain syndrome, the iliotibial band friction syndrome, and patellar tendinopathy. All recorded injuries could be attributed to overuse. At this point, it is worth mentioning that the national healthcare system allows patients on the one hand to self-report to a medical specialist without previous consultation with a gatekeeper (general practitioner, physiotherapist) or into a hospital, where the vast majority of trauma injuries will be treated. Consequently, the present study includes data from minor to severe overuse injuries in a heterogeneous study population.

The presentation of injured runners over the months shows a normal distribution with a peak in March. An increase was observed during the early winter months. This observation coincides with the onset of the “Vienna City Marathon”, the largest national running event with more than 40,000 participants, which is held on the second weekend in April each year. In contrast, a retrospective study from Vancouver reported a decrease over early winter months, following a reduction during Christmas holidays and an injury increase in marathon runners during months prior to the largest marathon in the area [11]. As previously reported [20], race preparation is related to a higher risk of sustaining a RRI. While there is no evidence that participating in long-distance races is associated with overuse injuries, presented data suggest that the accompanying increase in training load must occur incrementally and include regeneration periods.

### 4.2. Anthropometric and Personal Profile Data

#### 4.2.1. Sex

A possible sex discrepancy (56% female, 44% male, ratio female:male: 1.25) was observed in this study indicating either a generally higher number of female runners or women consulting a medical doctor sooner or a higher injury rate in women than men. While earlier studies showed lower ratios (0.76) [11], more recent literature shows a higher incidence in female injured runners (1.16) [5] suggesting that running is nowadays not predominantly a male sport. Data from a nationwide survey have reported a ratio over the past five years between female and male Austrians, who run at least once a week to be 0.78 (range: 0.65–0.89). The “Vienna City Marathon” showed a finisher ratio of 0.52 (range 0.48–0.55) for the half- and 0.24 (range 0.22–0.28) for the full marathon (both increasing) over the same period [21]. If the injury incidence across sexes was equal, the anticipated result would be a higher incidence of RRIs in men, especially across runners with higher total activity. Those runners are more likely to participate in long-distance races. However, the sex ratio in runners in the present study with five or more years of experience was 1.04. To investigate if there was a difference in visit rates to doctors among sexes, the patients were asked to indicate how often they visited a doctor per year. Women reported 2.6 visits, which is 0.35 times more than men, which corresponds to the well documented average yearly gynecologist visit rate of 0.41 [22] and could explain a higher rate in medical visits in women. In summary, the female to male discrepancy seems to be based on a higher injury rate in women rather than on a higher number of female runners in general, higher participation rates in race events, or more frequent medical consultations in women.

There were no other significant links between sex and any injury or location in the presented population. Unfortunately, very few studies have differentiated between sex and injury locations/diagnoses. Taunton et al. [5] reported higher injury rates in men for Achilles tendinopathy and gastrocnemius injuries and significantly higher prevalence in women for PFPS and ITBFS, which was also in accordance with data from Macintyre et al. [11]. The authors suggest that further studies should focus on specific injuries to clarify possible sex discrepancies and their causes.

#### 4.2.2. Height

Our sample showed on one hand significantly shorter patients in the PFPS group. Although patients with PFPS have reported lower body height in most studies, height did not attain statistical difference in any of them [23]. Since sex, as a categorical variable, has very little statistical power in the performed tests, it did not reveal any statistical significance. However, female patients were significantly smaller than male and the reported association could be regarded as an indicator for different sex, rather than actual height.

#### 4.2.3. Body Mass Index (BMI)

The ratio of bodyweight/height^2^ (BMI) was higher in patients with spinal/lower back injuries while lower BMI was observed in patients with knee injuries, ITBFS, and patellar tendinopathy. This relationship has also been previously reported [5]. On one hand, excessive biomechanical stresses act on the musculoskeletal system in individuals with higher BMI and a high percentage of body fat. On the other hand, it has been hypothesized that those with a low BMI may not have enough lean body mass to support their weight during the stresses of vigorous physical activity [24,25].

#### 4.2.4. Injury History

A prior history of injury was repeatedly reported to be associated with an increased risk of reinjury [15,20]. There are several plausible reasons for it: the actual cause for the pathology is still present, the repaired tissue is reduced in its function or protective role, or the injured tissue may not be healed completely. In the present study, 67.2% of the patients reported similar injuries in the past. Injury history contributed mainly to knee injuries and affected especially patients with ITBFS where 20 out of 21 patients presented a history of similar complaints. Hence, the return to running, especially in these patients, must be gradual and training distance and techniques that produce pain should be avoided.

### 4.3. Training Analysis

#### 4.3.1. Training History and Load

While higher overall training load seemed to be protective against PFPS and patellar tendinopathy, it was positively associated with injuries of the lower leg and ankle instability. Both observations have been documented in the literature. While a higher training load can reflect higher experience, less training errors, and more efficient anatomical adaptation, it can also result in a higher degree of fatigue and altered kinematics. These findings indicate that the risk of injury may not be simple and there may be a fine balance between overuse and underconditioning among runners.

#### 4.3.2. Stretching and Warming-Up Habits

Stretching after a run and warming up showed a negative association with hip/pelvis injuries. The benefit of stretching and warming up habits in runners shows mixed results in the literature [26,27,28]. However, the data in the present study does not include type, duration, and intensity of warming up and stretching and might be insufficient to deliver valid conclusions.

### 4.4. Footwear Analysis

Due to the minor response on the exact model the interpretation of the collected data concerning running shoes is limited. Analyzing the training data for the runners able to name their shoe model using an unpaired *t*-Test, the mean running history (6.6 vs. 7.3 years, *p* = 0.550), but also the complete training load was slightly higher compared to the rest of the study population; however, these differences were not statistically significant. Considering this and the low response rate itself, the choice of the optimal running shoe type is for many runners either not of great importance or there is simply a lack of awareness when purchasing footwear, especially in novice and less experienced runners.

Another interesting aspect is the low fraction of runners diagnosed with pes valgus wears motion-controlled TRS (Table 5). This is particularly exciting since the idea to prevent running-related injuries by aligning the body’s skeleton and correcting excessive movements has been discussed controversially in recent years [19,29]. The relatively low proportion of runners wearing motion-controlled TRS, even though they are part of the group with pes valgus, which was perceived as a high-risk factor for RRIs for a long time, might be the first results of the ongoing discussion. Future research will show if the paradigm of aligning by specific shoes will change and furthermore if this scientific knowledge will be relevant for the buying decision of regular runners.

### 4.5. Anatomic Malalignment Analysis

#### 4.5.1. Spine and Pelvic Malalignment

High-impact loads generated during running are distributed through the lower extremity kinetic chain and converge in the lumbar spine. Any changes in this chain will lead to compensatory changes in the involved segment as well as other segments in its kinetic chain [30]. Accordingly, scoliosis was one of the most meaningful positive associated factors for lower back injuries, but also hip and pelvis injuries. Leg length inequality can create a pelvic tilt, secondary scoliosis, and increased activity in lumbar musculature due to a lateral shift. Pelvic obliquity was observed in a large number of patients (Figure 3). Interestingly, it was not associated with any specific injury. In further research authors recommend differentiation between structural and functional scoliosis to fully understand the aetiology of running-related spine injuries.

#### 4.5.2. Knee Malalignment

In the present study, knee malalignment included all anomalies of the patella, ligamentous or general instabilities, and shortening of the thigh muscles. Present knee malalignments were associated with knee injuries, PFPS, and patellar tendinopathy with OR 3.8, 3.1, and 3.8, respectively. Anatomic and biomechanical conditions may predispose a mistracking of the patella resulting in unbalanced pressure distribution in the patellofemoral complex. This explains the positive association of knee malalignment with injuries in the patellofemoral joint. A lower training load will likely result in weaker active and passive stabilizers of the patellofemoral complex [31] exposing the runner to a higher risk of suffering PFPS.

#### 4.5.3. Leg Axis Malalignment

Previously, the varus/valgus knee has not been shown to be a risk factor for running injuries, despite its high incidence [5,10]. Similar to data from more than 2000 injured runners [5], we observed the highest incidence of the varus knee in patients diagnosed with knee injuries. The varus knee alignment induces higher stresses in the lateral musculature and stresses in the medial aspect of the knee as well as the mistracking of the patella, which would coincide with the reported pathologies.

#### 4.5.4. Foot Malalignment

Excessive foot pronation and associated higher lower limb rotation in runners with pes planus is considered to be associated with PFPS [32] and stress fractures [33]. Present data revealed a correlation with present pes planus deformity and knee injuries (OR 3.59). Taunton et al. [5] showed similarly the highest incidence of pes planus in patients with patellar tendinopathy, PFPS, tibial stress syndrome, and ITBFS. Similarly, Kuhman et al. [34] found the peak eversion angle is associated with injury in cross-country runners. Interestingly pes planus was protective of foot/ankle injuries (OR 0.35). As expected, pes cavus was associated with ankle instability.

### 4.6. Limitations and Protective Factors

Three limitations of this study remain to be mentioned. First, although data collection was conducted over a three-year period, a much larger population would have been necessary to evaluate the associated factors for all diagnosed injuries. In order to carry out a robust analysis with adequate sample size, regression analysis was limited to the seven most commonly found anatomic malalignments (96.5% of all malalignments) and the five most common specific diagnoses (42.2% of all diagnoses). Second, data collection could not be performed in a controlled manner. Instead, the authors had to rely on partially self-reported patient data. By asking participants very specific questions and offering help texts, it was aimed for accuracy maximization of the answers and estimations in participants’ self-reports. Nevertheless, due to the nature of the data collection instrument a general risk of a certain level of under- and over-reporting remains. Furthermore, we were unable to capture all complex mechanisms potentially causing an individual runner to suffer a running-related injury. Therefore, our data may only explain a certain aspect of much wider, underlying variables influencing a runner’s risk for an overuse injury. Furthermore, certain training data, such as ’participation in alternative sports´ or ´running underground´ were too unspecific and could only be presented descriptively. Third, the presented model was constructed from possible contributing factors using a dependent variable of runners with a specific injury compared with a control group of runners who experienced a different injury. A limitation of the external validity of such regression modeling is the lack of differentiation between injured and non-injured runners. Nevertheless, the recruitment of non-injured runners is only feasible in selected populations and with prospective study design. In order to conduct a prospective study, one would have to take into account a selected and biased study population. Finally, the presented approach proved valid after most findings and the magnitude of statistical significances were plausible and coincided with data from previously published studies.

Finally, presented odds ratios (see Table 7) significantly different from 1 which may be caused by a risk or a protective effect for a specific injury as well as by the contrary effect against one of the other injuries. As a result of this logistic, regression models revealed a series of factors with odds ratios <1. Those are for instance malalignments of the knee or varus knee negatively associated with lower leg and foot/ankle or lower back (spinal injuries), respectively. In theory, those variables could also represent a protective factor for the specific injury or location. However, the more reasonable interpretation is the contrary effect caused by the highly associated knee malalignments and knee injuries. Although not further interpreted, the authors decided to display these results for purposes of data integrity and good scientific practice.

## 5. Conclusions

In conclusion, running-related injuries are multifactorial, associated with personal data, training load, anatomic malalignments, and injury history. While it is impossible to define a one-fits-all formula to reduce the risk for RRI in general, runners at a high risk of a specific injury could be identified based on patient-specific training profile and running gait as well as on pre-existing malalignments, such as scoliosis, patellar squinting, knee malalignments, and/or varus knee and help to induce an appropriate and balanced training adaptation. Furthermore, awareness of injury risks and prevention should be raised in running schools and by medical specialists. The authors strongly recommend further studies focused on specific injuries in combination with related malalignments as well as on detailed training habits, rather than research on running-related injuries as a generalized pathology.

In the end, running has remained a popular activity over decades even though it has been associated with a high incidence of overuse injuries. By presenting the injuries the authors did not aim to discourage from running but to provide data for better understanding and contribute to their prevention. We believe running is still beneficial in many respects, enriching, and simply fun.

## Figures and Tables

**Figure 1 jcm-09-00438-f001:**
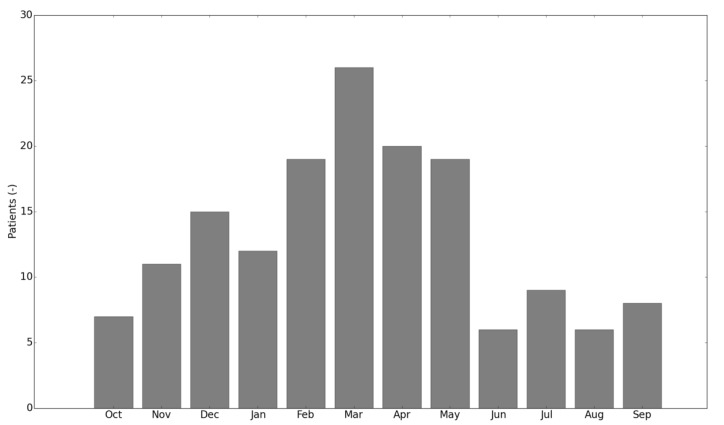
Months of presentation.

**Figure 2 jcm-09-00438-f002:**
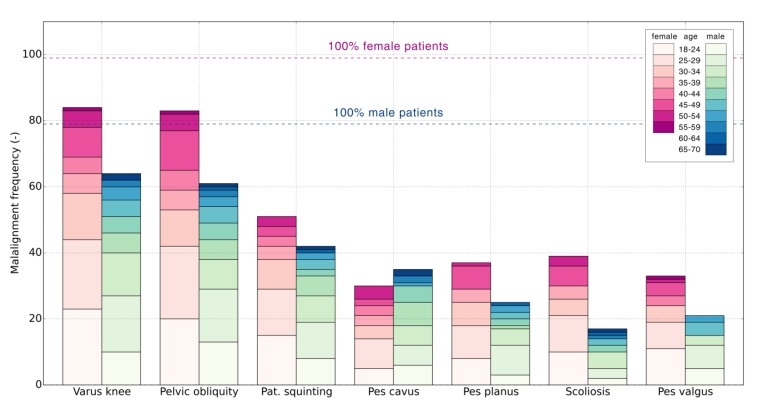
Distribution of most common malalignment across sexes and age groups. Dashed lines represent 100% of all female (99) and male (79) patients.

**Figure 3 jcm-09-00438-f003:**
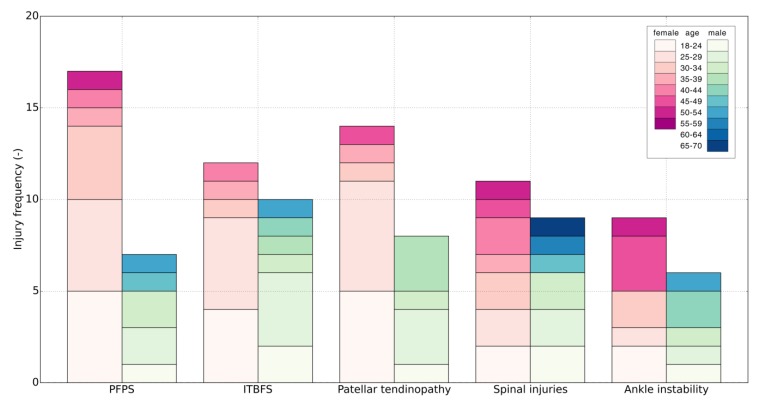
Distribution of most common running-related injuries (RRIs) across sexes and age groups.

**Table 1 jcm-09-00438-t001:** Descriptive information for currently largest available studies on running-related injuries.

Reference	Data Collection Period	Country	Number of Patients	Description of Population
Jacobs and Berson [7]	1984	USA	210	participants of specific race
Marti et al. [15]	1984	Switzerland	1994	male participants of specific race
Macintyre et al. [11]	1980–1985	Canada	4173	all
Wen et al. [10]	not specified	USA	255	participants of specific training
Taunton et al. [5]	1998–2000	Canada	2002	all
Taunton et al. [9]	1996–2000	Canada	844	participants of specific training
Lun et al. [2]	not specified	Canada	87	ruining duration > 20 km/week
Rauh et al. [6]	1996	USA	421	high school cross-country runners
Van Middelkoop et al. [17]	2005	The Netherlands	195	participants of specific race
Buist et al. [16]	2008	The Netherlands	163	participants of specific training
Rasmussen et al. [14]	2011	Denmark	68	participants of specific race
Nielsen et al. [12,13]	2011–2012	Denmark	254	novice runners
Kluitenberg et al. [8]	2013	The Netherlands	185	participants of specific training

**Table 2 jcm-09-00438-t002:** Anthropometric and personal profile characteristics and differences between sexes of the study population.

Patients’ Characteristics	Total	Range	Missing Values (%)	Female	Male	*p*-Value
Number (*n* (%))	178 (100%)	*n*/a	0	99 (55.6%)	79 (44.3%)	0.154
Age (years)	33.3 ± 11	18–69	0	32.5 ± 10.3	34.4 ± 11.6	0.241
Height (cm)	173.6 ± 8.9	150–200	5	167.8 ± 5.6	180.7 ± 6.7	<0.001
Weight (kg)	68.8 ± 12.5	42–105	2.2	61.2 ± 8.6	78.9 ± 9.3	<0.001
BMI (kg/m^2^)	22.8 ± 2.6	16.7–30.8	7.3	21.8 ± 2.4	24.1 ± 2.3	<0.001
Education (CE/HS/PTU/BA/MA/PhD (%))	(3.4/26.4/4.5/14/40.4/6.2)	*n*/a	5.1	(4.1/33/4.1/16.5/41.2/1)	(2.8/20.8/5.6/12.5/44.4/13.9)	0.013

Abbreviations: CE (compulsory education), HS (high school), PTU (vocational school), BA (bachelor’s degree), MA (master’s degree), PhD (doctoral degree).

**Table 3 jcm-09-00438-t003:** Runner’s profile data of the study population.

Training Profile	Total	Missing Values (%)	Female	Male	*p*-Value
Running history (years)	6.9 ± 7.7	0.5	5.3 ± 4.5	9 ± 10.1	0.069
Training duration (hours/week)	2.8 ± 1.5	4.4	2.6 ± 1.3	3.1 ± 1.7	0.071
Mileage (km/week)	25.4 ± 14.9	15.1	23.9 ± 13.5	27.2 ± 16.5	0.199
Frequency (runs/week)	3 ± 1.1	2.8	3.0 ± 1.1	3.1 ± 1.2	0.750
Pace (km/h)	9.3 ± 1.9	19.6	8.8 ± 1.6	9.9 ± 2.1	<0.001

**Table 4 jcm-09-00438-t004:** Shoe categories overview.

Shoe Category *n* (%)	Female	Male	Total
Trail running	0 (0.0%)	2 (5.4%)	2 (2.4%)
TRS (neutral)	27 (56.3%)	16 (43.2%)	43 (36.5%)
TRS (motion-controlled)	16 (33.3%)	15 (40.5%)	31 (50.6%)
Racing flats	0 (0.0%)	0 (0%)	0 (0.0%)
Barefoot shoes/MRS	5 (10.4%)	4 (10.8%)	9 (10.6%)
Total	48 (100%)	37 (100%)	85 (100%)

Abbreviations: TRS (traditional running shoes), MRS (minimalist running shoes).

**Table 5 jcm-09-00438-t005:** Relationship between the malalignment pes valgus and the use of motion-controlled traditional running shoes (TRS).

Pes Valgus	TRS (Motion-Controlled)		Total
	No	Yes	
**No**	45 (50.0%)	22 (24.4%)	67 (74.4%)
**Yes**	14 (15.5%)	9 (10.0%)	23 (25.5%)
**Total**	59 (65.5%)	31 (34.4%)	90 (100%)

Abbreviations: TRS (traditional running shoes).

**Table 6 jcm-09-00438-t006:** Injury locations across sexes.

Location	Female	Male	Total
Foot	15 (13.0%)	6 (7.1%)	21 (10.6%)
Ankle	17 (14.8%)	13 (15.5%)	30 (15.0%)
Calf/Achilles	5 (4.3%)	10 (11.9%)	15 (7.5%)
Lower Leg	5 (4.3%)	6 (7.1%)	11 (5.5%)
Knee	48 (41.7%)	34 (40.5%)	82 (41.2%)
Upper Leg	3 (2.6%)	0 (0.0%)	3 (1.5%)
Hip/Pelvis	11 (9.6%)	6 (7.1%)	17 (8.5%)
Lower Back	11 (9.6%)	9 (10.7%)	20 (10.1%)
Total injuries	115(100%)	84 (100%)	199 (100%)

**Table 7 jcm-09-00438-t007:** Univariate logistic regression analyses for the five most common injury locations and five most common injuries.

Location	Injury	Associated Variables	*p*-Value	Mean ± SD	OR	95% CI
Knee		Younger age	0.001	30.44 ± 8.35	1.050	1.01–1.08
		Lower BMI	<0.001	22.08 ± 2.34	1.249	1.09–1.42
		Previous injury	0.041	-	2.002	1.02–3.98
		Lower running history	0.079	5.83 ± 5.86	1.038	0.99–1.08
		Varus knee	0.001	-	4.118	1.67–11.81
		Patellar squinting	<0.001	-	3.081	1.67–5.78
		Pes planus	0.044	-	1.894	1.01–3.56
		Knee malalignment	<0.001	-	3.871	2.07–7.40
	PFPS	Lower height	0.022	169.67± 9.04	1.065	1.00–1.12
		Less weekly hours	0.029	2.00 ± 0.81	2.072	0.57–1.42
		Lower mileage	0.014	17.90 ± 9.13	1.065	1.01–1.12
		Lower frequency	0.047	2.42 ± 0.88	2.026	1.24–3.30
		Lower pace	0.059	8.47 ± 1.76	1,281	0.99–1.65
		Hip obliquity	0.076	-	0.406	0.15–1.10
		Patellar squinting	0.017	-	3.092	1.21–9.05
		Knee malalignment	0.008	-	3.601	1.35–11.45
	ITBFS	Lower BMI	0.016	21.55 ± 2.54	1.261	1.04–1.52
		Previous injury	0.001	-	10.193	2.03–248.19
	Patellar tendinopathy	Younger age	0.032	28.55 ± 6.95	1.060	1.00–1.11
		Lower BMI	0.074	21.82 ± 2.41	1.189	0.98–1.44
		Lower pace	0.077	8.47 ± 2.22	1.274	0.97–1.66
		Scoliosis	0.012	-	0.203	0.02–0.73
		Hip malalignment	0.094	-	0.408	0.15–1.17
		Knee malalignment	0.022	-	3.146	1.16–10.16
Lower leg		More weekly hours	0.015	3.60 ± 1.52	1.367	1.06–1.76
		Higher mileage	0.002	34.58 ± 20.68	1.044	1.01–1.07
		Higher frequency	0.035	3.48 ± 1.22	1.443	1.02–2.03
		Higher pace	0.036	10.35 ± 2.02	1.459	1.13–1.88
		Male sex	0.062	-	2.238	0.95–5.46
		Patellar squinting	0.005	-	0.288	0.10–0.70
		Knee malalignment	0.002	-	0.259	0.09–0.63
Foot/Ankle		Previous injury	0.044	-	0.483	0.23–0.98
		Patellar squinting	0.001	-	0.331	0.16–0.65
		Pes planus	0.075	-	0.518	0.23–1.06
		Knee malalignment	<0.001	-	0.291	0.14–0.57
	Ankle instability	More weekly hours	0.043	3.68 ± 2.39	1.352	1.00–1.81
		Pes cavus	0.060	-	2.825	0.95–9.00
Hip/Pelvis		Higher age	0.072	37.94 ± 15.46	1.039	0.99–1.08
		Higher running history	0.042	10.78 ± 13.47	1.053	1.00–1.10
		Scoliosis	0.016	-	3.520	1.25–10.41
		Stretching after run	0.010	-	0.228	0.07–0.69
Lower back	(Spinal injuries)	Higher BMI	0.012	24.35 ± 2.74	1.28	1.05–1.55
		Scoliosis	<0.001	-	8.282	2.97–27.40
		Varus knee	0.001	-	0.190	0.06–0.52
		Warm-up	0.043	-	0.366	0.12–0.97

Analysis for the metric values includes the level of statistical significance (*p*-value), mean ± standard deviation (SD) in the group with this injury as well as the odds ratio (OR) and the corresponding 95%-confidence interval (CI). Analysis of nominal values includes the p-value, the odds ratio (OR) and the corresponding 95%-confidence interval (CI). Abbreviations: iliotibial band friction syndrome (ITBFS), patellofemoral pain syndrome (PFPS).

**Table 8 jcm-09-00438-t008:** Multivariate logistic regression analyses for the five most common injury locations and five most common injuries.

Location	Injury	Associated Variables	*p*-Value	OR	95% CI
Knee		Lower BMI	0.034 *	1.177	1.01–1.37
		Varus knee	0.004 **	7.380	1.86–29.21
		Pes planus	0.003 **	3.590	1.53–8.40
		Previous injury	0.020 *	2.671	1.16–6.13
		Knee malalignment	<0.001 ***	3.871	2.07–7.40
	PFPS	Lower height	0.038 *	1.062	1.00–1.12
		Lower frequency	0.011 *	1.988	1.16–3.37
		Knee malalignment	0.039 *	3.104	1.05–9.12
	ITBFS	Lower BMI	0.058	1.019	0.99–1.04
		Previous injury	0.004 **	1.181	1.18–1.32
	Patellar tendinopathy	Lower BMI	0.053	1.269	0.99–1.61
		Lower pace	0.010 *	1.272	0.94–1.70
		Hip malalignment	0.025 *	0.239	0.06–0.83
		Knee malalignment	0.056	3.859	0.96–15.46
Lower leg		Higher mileage	0.154	1.024	0.99–1.05
		Higher pace	0.065	1.332	0.98–1.80
		Knee malalignment	0.008 **	0.256	0.09–0.70
Foot/Ankle		Previous injury	0.064	0.486	0.22–1.04
		Pes planus	0.015 *	0.356	0.15–0.82
		Knee malalignment	<0.001 ***	0.263	0.12–0.56
	Ankle instability	More weekly hours	0.016 *	1.487	1.07–2.05
		Pes cavus	0.038 *	3.555	1.07–11.7
Hip/Pelvis		Longer running history	0.028 *	1.067	1.07–1.13
		Scoliosis	0.022 *	3.554	1.19–10.56
		Patellar squinting	0.037 *	3.750	1.08–13.00
		Stretching after run	0.007 **	0.204	0.06–0.65
Lower back	(Spinal injuries)	Higher BMI	0.041 *	1.273	1.01–1.60
		Scoliosis	<0.001 ***	12.669	3.27–49.03
		Varus knee	<0.001 ***	0.098	0.02–0.37

Analysis for all values includes the level of statistical significance (*p*-value), the odds ratio (OR) and the corresponding 95%-confidence interval (CI). Variables reaching various levels of significances were emphasized with asterisks (* *p* ≤ 0.05, ** *p* ≤ 0.01, *** *p* ≤ 0.001). Abbreviations: iliotibial band friction syndrome (ITBFS), patellofemoral pain syndrome (PFPS).

**Table 9 jcm-09-00438-t009:** Model equations for the most common injury locations and specific injuries.

Model	Equation
Knee injury	Log(odds) = 0.01 −0.16 × (BMI) + 0.98 × (injury history) + 2 × (varus knee) + 1.47 × (knee malalignment) + 1.28 × (pes planus)
PFPS	Log(odds) = 9.86 − 0.06 × (body height) − 0.69 × (weekly frequency) + 1.13 × (knee malalignment)
ITBFS	Log(odds) = 0.47 − 0.02 × (BMI) + 0.17 × (injury history)
Patellar tendinopathy	Log(odds) = 5.56 − 0.024 × (BMI) − 0.24 × (pace) − 1.43 × (hip malalignment) + 1.35 × (knee malalignment)
Lower leg	Log(odds) = −4.37 + 0.02 × (weekly mileage) + 0.29 × (running pace) − 1.36 × (knee malalignment)
Foot/Ankle	Log(odds) = 1.21 − 0.72 × (injury history) − 1.34 × (knee malalignment) − 1.03 × (pes planus)
Ankle instability	Log(odds) = −3.11 + 0.40 × (weekly hours) + 1.27 × (pes cavus)
Hip/Pelvis	Log(odds) = −2.99 + 0.06 × (running history) + 1.27 × (scoliosis) − 1.59 × (stretching after run)
Lower back	Log(odds) = −7.42 + 0.24 × (BMI) + 2.54 × (scoliosis) − 2.32 × (varus knee)

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
