# Peer review of "Analysis of Running-Related Injuries: The Vienna Study"

_jcm, 2020, doi:10.3390/jcm9020438_

Round 1
Reviewer 1 Report
Thank you for the response letter and the clarification of my questions. The author’s have made several substantial changes to the methods and manuscript, which have improved the quality I think.
I still have several comments.
Introduction:
Line 42: place RRI between ( ).
Line 45: RROI is mentioned after overuse, but is a abbreviation for Running Related Overuse Injuries.
Line 46: While these injuries.... Do you refer to reference 6? Otherwise, rewrite the sentence to make clear what injuries you mean with ‘ these injuries’
In lines 48 to 69 you switch from RROI to RRI to running related injuries and back. Please check wether you have used the correct abbreviations everywhere. And use the abbreviations.
I don’t understand aim 3: investigate if different running footwear categories modify running incidences? Actually, I do understand the aim, but not how you could analyse this aim with your data. In the results section, some information on footwear is presented, but with this information, the aim cannot be investigated/analysed. Please explain how you would analyse aim 3.
Methods
Line 85-86: ‘patients do not have to always seek’ or should it be ‘do not always have to seek’ The latter sounds better to me, but I am not a native speaker.
Results
Line 183: The participants > replace with patients.
The results referring to aim 2 are presented after the results referring to aim 3.
Table 7: uniform the presented p-values, most common is 3 decimals, e.g. p = 0.004.
The same goes for the OR and CI. For the OR you use three decimals, for the CI only 2, but every now and than 3.
The CI for ITBFS and previous injury is very wide, and probably not trust worthy. The same goes for lower back and scoliosis.
The results for lower back and spinal injuries are the same, probably the same patients. If you place spinal injuries between ( ) after lower back, you can remove the spinal injuries results from the table.
Did you consider to include the scale variables, e.g. age, height or BMI, as categorical variables in the logistic regression analyses? To give more insight in the results (what is a lower BMI for example?)
Lower/higher running history: suggestion > longer and shorter?
Table 8: check decimals for p-values, OR and CI.
Remove results spinal injuries
Only present, and discuss the significant results in table 8.
Line 282: Hip/Pelvis patients showed a positive.. Insert association.
As far as I know, it is not usual to present a table like table 9.
Discussion:
The discussion is lengthy, but interesting to read.
Check the language, e.g. line 356: the authors suggest THAT further studies SHOULD.
Line 362 PFPS is mentioned twice, is this correct?
Check the discussion section after you removes the non significant results in table 8.
Reviewer 2 Report
I would like to thank the authors for a thorough revision of their manuscript. I would like to apologize for the length of time it took me to revise the second version of this manuscript. Here are specific comments.
L73 - Your aim that the study is to “present a heterogeneous running population from a Central Euro capital” is not clear. What is that purpose exactly? How is it assessed statistically? What’s the hypothesis? Do you mean you want to present injury and training characteristics of a Central Euro capital? That might be more clear. If that’s the case, then some of the data included in the results make more sense (see below).
L129-130 – In response to your revisions. (R2.10) The statistical analyses and group comparisons are certainly odd/unusual (comparing runners with specific injuries vs runners with other injuries). Why would characteristics of a runner with one injury be “better” than characteristics of runners with another injury? I’m not sure why this comparison is useful.
Authors’ Response
As discussed in Response to R2.7 the selection of the control was based on both, generally high
incidences and high incidence rates. Therefore, it again seemed plausible and legitimate to
compare runners, who suffered a specific injury with other runners, who suffered any other
injury. Furthermore, this approach has been used in previously published studies (e.g.
http://dx.doi.org/10.1136/bjsm.36.2.95). See also response to R2.7.
Actions Some of these aspects were emphasized in the paper.
- I should have made my original comment more clear. It is not clear why these statistical comparisons were made given your study purposes. Your statistical analyses should be expected based on a clearly defined purpose with a clearly defined hypothesis.
- Results section. I still have the same comment as in the original review. This paper is about injuries but there are many results regarding characteristics of runners and correlations among these characteristics. Again, I strongly recommend you tightly focus your results on your purposes and hypotheses. Two of the three aims are fairly clear. The first (added) aim is not clear (see comment above). You also need some hypotheses to go along with these aims. Everything is Table 7 is great, but focus your results section on this and not on unrelated correlations (e.g., “positive correlation between mileage and hour…”. If you clarify purpose 1 it might help.
(R2.16) If specific hypotheses were identified, the discussion could be focused more specifically on addressing those hypotheses.
Authors’ Response
See response to R2.11 and R2.15.
Actions See action to R2.11 and R2.15.
- You still do not have any hypotheses, only aims/purposes. Please address this.
L290 – see comment above about making the first aim more clear
L295 – “high running history”. What is meant by this?
L297-304 – these are method and results, mostly. They could be omitted from this section of the discussion or discussed in specific sections within the discussion. They seem oddly placed at the start of the discussion. As mentioned in the previous revisions, I recommend you formulate hypotheses and then address each hypothesis within the discussion section, one by one.
L400-402: do you mean “more training experience” might mean less training errors? Many people who have high training loads make training errors. What do you mean by “more efficient anatomical adaptation”? And why is that more efficient with higher training load? It all depends. Some references are needed for these statements and they need to be clarified.
Some instances you spell out running-related overuse injuries and other instances you use RROI. I recommend you choose one and use it throughout.
L453-455: But varus knee position also increases contact area on medial aspect of knee…
L456: Foot Malalignment: see recent prospective analyses on foot pronation and injury in cross-country runners:
Kuhman et al., Comparison of ankle kinematics and ground reaction forces between prospectively injured and uninjured collegiate cross country runners. Human movement science, 2016.
Dudley et al. A Prospective Comparison of Lower Extremity Kinematics and Kinetics Between Injured and Non-Injured Collegiate Cross Country Runners. Human movement science, 2017.
A discussion of the multivariate analysis is highly recommended (table 8). In my opinion, that’s the most valuable aspect of the study and will greatly add to the value of the discussion section.
Round 2
Reviewer 1 Report
Thank you for the revised version of this manuscript.
I have read the reply to my comments and the article, and have some minor points:
Lines 223-224 are a duplicate of lines 213-214.
In my last review I suggested to describe the signifiant results only (in table 8). In Table 7 you show all significant (p<0.05) relationships, revealed from a univariate model, which could be found in the presented sample. However, variables that showed at least a trend (p<0.1) in the univariable model were combined in a multivariable regression model. Therefore in this table 7, the level of significance should be set at p<0.1 instead of p<0.05.
In table 8, the results of the mutivariable model are presented. Some associations are still significant, some now show a trend (the CI include 1) The iliotibial band friction syndrome for example showed a positive association with previous injury and a negative association with BMI However, the latter association was not significant. The same goes for results for Patellar tendinopathy, lower leg, foot/ankle. Please check your table and the accompanying text in the result section and discussion.
Author Response
Please see the attachment.

This manuscript is a resubmission of an earlier submission. The following is a list of the peer review reports and author responses from that submission.
Round 1
Reviewer 1 Report
Research in the field of running and running related injuries is, to my opinion, always important, as running remains a very popular sport, the incidence rates are still very high, and more information on injury prevention is necessary. I do have several questions/remarks for the authors.
Abstract: 37 contributing factors is misleading, as some of them are related to more than one injury of injury location.
Did you control the univariate analyses for confounding or effect modification? Why did you not use multivariable regression analyses? The t-test are not mentioned in the abstract. Further more, as far as I know, a T-test is used to describe differences between the mean of two groups, and not for the association of two factors.
Introduction: Are you sure that all references mentioned refer to Running related overuse injuries, or are acute injuries also included in some of them? I miss a reference after high treatment costs (line 44).
Your remark in line 64 seems to be conflicting with line 59. "Finally, the most extensive study with over 2,000 patients proposed the necessity for a precise measure of weekly running distance an running experience to validate existing data. Therefore a detailed up-to date retrospective study.. I think a prospective study is necessary for precise measurements.
Methods: The population of runners is described as heterogeneous. I doubt whether these runners are truly heterogeneous as they all visited an orthopedics practice. For me this is an indication of having a sever RROI. In our country you don’t visit an orthopedic center or surgeon with a minor injury. If you do have a minor injury, you might visit a physiotherapist or general practitioner. Therefore, I think your patients might be heterogeneous with regards to personal characteristics, but not with regard to the injuries. Is this a correct assumption?
Furthermore, due to your exclusion criteria in line 86 and 87, you will exclude part of the novice runners.
According to your definition in lines 83-86, an injury could be acute as well, not only Overuse. I would mention the onset of the injury as well. That only overuse injuries were seen, doesn’t mean they don’t exist.
A lot of RRI are not medically treated in our country. I think this is the case for running injuries in every country. Hence, this is another indication that ‘ your’ injuries are more sever, and that your conclusions are not applicable for all RRIs.
Again, t-test are not used for associations, but to indicate differences between two groups. Line 103. Why didn’t you use logistic regression analyses for these variables as well? Your outcome variable remains the same. And using logistic regression analyses for all variables makes it easier to enter the significant variables into a multi variable model to indicate the most important variables. For example, in table 7, both lower weekly hours, lower mileage, and lower frequency are associated with PFPS. Without doubt these three variables are related with each other as well.
The authors made the assumption that based on high incidences and incidence rates, it is plausible to compare injured runners with injured runners. Again, incidence rates are high, but not all runners seek medical attention (See Kemler, Blokland, Backx and Huisstede, 2018 for more information on medical treatment in the Netherlands. Possibly more studies with information on medical treatment are available). This should be taken into account for the conclusions of this study.
Results: The results are described extensively. I question the added value of the information on footwear (whether runners know the brand and specific model of their shoes). This could be removed. The percentage that have changed their shoes within 30 days prior to the injury might be of interest. As only 45.5% was able to name the specific model of their shoes, what is the value of the information in table 4. Did you verify the information the runners gave you about their shoes?
A lot of results are presented in figures and tables? How much figures and tables are allowed in this journal?
I think the results in line 207-209 are not surprising. But if their is such a high correlation, why did you still use all of them in the logistic analyses. Why not pick one of them?
Table: remove predictor in the table head. The variables are associated, but they are not a predictor, as this is a retrospective study. Cause and consequence are not clear in these analyses. For example, is a low mileage a cause of PFPS, or is this an injury with a gradual onset, which causes runners to slow down and to decrease their training load? Hard to say.
Again, I wonder whether all association will hold with multivariable regression analyses.
Discussion:
Although the discussion section is very lengthy already, add a section in which you discuss the nature (severity) of the injuries included and the usability of your results for all RRIs.
In line 324-330 you discuss the data concerning running shoes. And you question whether the choice for an optimal running shoe is for many runners not of great importance, or there is a simply lack of awareness, especially in novice and less experienced runners? What is your definition of novice and less experienced runners? You haven’t mentioned them before. Did you perform sub analyses?
Conclusions:
Your conclusions should be related to more severe rroi, not all running related injuries.
Finally, I wonder who could make use of the data of your study. Ordinary runners won’t get a running/training profile when they start running. So how do we identify these high risk runners? Are the results only of use for the professional runners?
Reviewer 2 Report
This study aimed to provide an extensive and up-to-date analysis of RROI and analyze a broad range of contributing factors. There is value to this investigation but there are multiple issues with the writing (both grammatical and organizational) and some of the interpretations are perhaps slightly strong and need to be toned down. Some important literature on this topic is lacking in the introduction and discussion which limits in the interpretations. The major limitation of this study is the retrospective design that limits the identification of true risk factors for RROI. Also, the statistical analyses are unusual and potentially problematic. I have provided specific comments below.
Specific Comments
Abstract
In abstract it is unclear if running injuries were retrospective or prospective.
L28 – anatomic malalignments is mentioned in conclusion but not in results.
Introduction
L50 – there have been dozens of epidemiological studies investigating risk factors for RRI in the last 5 years. See work from RunSafer (Rasmus Nielsen) in Denmark.
L56 – training does fall in the category of extrinsic risk factors
L59 – To what data are you referring here?
L57-58 – This sentence regarding footwear seems to be unrelated to the rest of the introduction. Perhaps more info on running footwear and injury (although evidence is ample and not showing that shoes contribute to injury development in runners).
In general, retrospective studies are not ideal to identify RROI risk factors. Prospective and follow-up studies are more well-suited for such purposes.
It would be useful to include hypotheses.
Methods
What are malalignments specifically? How were those diagnosed/measured? It’s not clear in the manuscript.
A fair amount of detail is missing from methods, especially regarding specific how the data were collected (scales, categorical data?)
The statistical analyses and group comparisons are certainly odd/unusual (comparing runners with specific injuries vs runners with other injuries). Why would characteristics of a runner with one injury be “better” than characteristics of runners with another injury? I’m not sure why this comparison is useful.
Results
The results are slightly confusing and disorganized at times. The purpose of the study is to identify risk factors RROI, but there are many results unrelated to this like comparing men and women runners. I recommend the authors focus on the injury data and comparing groups (injured vs controls).
Same comment regarding malalignment, no context provided for what those are.
Section 3.6 is the most valuable and relevant to the purpose of the study.
All tables should be in the same format. Table 7 (despite not having a title) is the best formatted one out of the lot.
Discussion
The discussion section needs a lot of work. It’s difficult to follow all the different paragraphs. I recommend focusing the discussion on the purpose/hypotheses of the study more directly and clearly. Since there are many results, doing this will improve the clarity of the discussion.
If specific hypotheses were identified, the discussion could be focused more specifically on addressing those hypotheses.